# High Rates of Multidrug-Resistant *Escherichia coli* in Great Cormorants *(Phalacrocorax carbo*) of the German Baltic and North Sea Coasts: Indication of Environmental Contamination and a Potential Public Health Risk

**DOI:** 10.3390/pathogens11080836

**Published:** 2022-07-27

**Authors:** Stephanie Gross, Anja Müller, Diana Seinige, Manuela Oliveira, Dieter Steinhagen, Ursula Siebert, Corinna Kehrenberg

**Affiliations:** 1Institute for Terrestrial and Aquatic Wildlife Research, University of Veterinary Medicine Hannover (TiHo) Foundation, 25761 Büsum, Germany; stephanie.gross@tiho-hannover.de; 2Institute for Veterinary Food Science, Justus Liebig University Giessen, 35392 Giessen, Germany; anja.mueller@vetmed.uni-giessen.de (A.M.); corinna.kehrenberg@vetmed.uni-giessen.de (C.K.); 3Office for Veterinary Affairs and Consumer Protection, Ministry of Lower Saxony for Food, Agriculture and Consumer Protection, 29221 Celle, Germany; diana.seinige@gmx.de; 4CIISA—Centre for Interdisciplinary Research in Animal Health, Faculty of Veterinary Medicine, University of Lisbon, 1300-477 Lisbon, Portugal; moliveira@fmv.ulisboa.pt; 5Associate Laboratory for Animal and Veterinary Sciences (AL4AnimalS), Avenida da Universidade Técnica, 1300-477 Lisbon, Portugal; 6Fish Disease Research Unit, University of Veterinary Medicine Hannover, 30559 Hannover, Germany; dieter.steinhagen@tiho-hannover.de

**Keywords:** antimicrobial resistance, Enterobacteriaceae, suliformes, wildlife, one health

## Abstract

Antimicrobial-resistant bacteria pose a serious global health risk for humans and animals, while the role of wildlife in the dynamic transmission processes of antimicrobial resistance in environmental settings is still unclear. This study determines the occurrence of antimicrobial-resistant *Escherichia coli* in the free-living great cormorants (*Phalacrocorax carbo*) of the North and Baltic Sea coasts of Schleswig-Holstein, Germany. For this, resistant *E.*
*coli* were isolated from cloacal or faecal swabs and their antimicrobial resistance pheno- and genotypes were investigated using disk diffusion tests and PCR assays. The isolates were further assigned to the four major phylogenetic groups, and their affiliation to avian pathogenic *E. coli* (APEC) was tested. Resistant *E. coli* were isolated from 66.7% of the 33 samples, and 48.9% of all the resistant isolates showed a multidrug resistance profile. No spatial differences were seen between the different sampling locations with regard to the occurrence of antimicrobial resistance or multidrug resistance. Most commonly, resistance percentages occurred against streptomycin, followed by tetracycline and sulfonamides. More than half of the isolates belonged to the phylogenetic group B1. Of all the isolates, 24.4% were classified as APEC isolates, of which almost 82% were identified as multidrug-resistant. These results add information on the dispersal of antimicrobial-resistant bacteria in wild birds in Germany, thereby allowing conclusions on the degree of environmental contamination and potential public health concerns.

## 1. Introduction

Antimicrobial resistance in bacteria is a global health threat in human and veterinary medicine [1,2]. The increase in the frequency of multidrug-resistant bacteria over the past decades has been driven by the selective pressure imposed by the use and misuse of antibiotics [1]. Antimicrobial-resistant bacteria are not limited to humans, companion animals, and livestock; they can be found in the environment naturally [3,4,5] as well as a consequence of anthropogenic activities [5,6,7,8]. Moreover, antimicrobial agents themselves have already been found in soil, surface, and ground water, and some of them have been shown to persist in the environment for months or even years [9,10,11,12]. In this context, wild animals can play an important role as reservoirs and vectors for antimicrobial-resistant bacteria. However, their importance in the dissemination of these bacteria and resistance genes is still incompletely explored [13]. 

Especially interesting are the aquatic ecosystems as they are subjected to high levels of different anthropogenic impacts [14]. Accordingly, various studies have detected antimicrobial-resistant bacteria in natural waterbodies, often finding evidence of anthropogenic causes, including sewage discharges, agricultural run-off, and health-care effluent [6,7,15,16,17,18,19,20]. Controversial results have been found with regard to whether wastewater treatment plants select for or against antimicrobial-resistance genes [21,22,23,24,25]. Furthermore, there are indications that wastewater treatment, although generally reducing the amount of bacteria, might select for multidrug resistance and might lead to an accumulation of resistance genes [26,27]. Genetic exchange occurs mainly via horizontal transfer of genetic material between bacteria including environmental bacteria and microbes that originated from humans and animals [28,29]. However, resistant bacteria were also detected in environments with an absence of external influences [30]. Here, birds may play an important role as bird faeces were found as another important contamination source for antimicrobial-resistant bacteria in natural waterbodies [31]. The birds themselves are likely to acquire the resistant bacteria from the environment [32], indicating the complex transmission dynamics of antimicrobial resistance in the environment.

Enterobacteriaceae from surface waters have been identified as important reservoirs of antimicrobial-resistant bacteria and resistance genes [33]. Within the family Enterobacteriaceae, *Escherichia coli* is of specific interest as it colonizes the intestines of mammals and birds. Here, it mainly acts commensally, but some strains are known pathogens, causing severe intestinal or extra-intestinal diseases in both humans and animals [34]. In addition, *E. coli* can easily receive, harbour, and pass antimicrobial resistance genes, mainly via horizontal gene transfer to other bacteria [35,36]. *E. coli* can be disseminated by faeces, manure, sewage sludge, or treated wastewater to the terrestrial and aquatic environment where it survives, depending on the existing abiotic and biotic factors, for varying periods of time [27,37]. A recent study demonstrated the correlation of the access to treated wastewater and the increased antimicrobial-resistance burden in wild birds [38]. In this context, migratory birds in particular may play an important role in the dissemination of antimicrobial-resistant bacteria over long distances and into pristine niches [32,39,40,41]. As humans, wildlife, and the environment share similar *E. coli* strains [42], resistant strains in the environment pose a potential infection source for humans [18,43]. Altogether, this highlights the potential of *E. coli* as an indicator of the spread of antimicrobial resistance in the environment as well as its importance as a public health concern.

Antimicrobial-resistant *E. coli* have been reported from various wild bird species [44,45,46,47]. For the present study, great cormorants (*Phalacrocorax carbo*) were selected as the study object for several reasons. First of all, large breeding colonies allow an easy access to faecal samples. Secondly, their feeding ecology is almost exclusively based on fish [48,49,50], making them perfect bioindicators for the occurrence of antimicrobial-resistant bacteria in the environment [51] as they rarely feed on anthropogenic waste. Thirdly, they have already been shown to frequently carry resistant *E. coli* isolates [40,52,53,54]. Lastly, due to migration, great cormorants might also be able to disseminate resistant bacteria over extended distances [40,55,56].

The aim of this study was to investigate the occurrence of antimicrobial-resistant *E. coli* in great cormorants in the German federal state of Schleswig-Holstein. Here, great cormorants were used as a potential sentinel aquatic bird species to understand whether there were regional patterns according to the origin of the animals. Moreover, the level of multidrug resistance among the isolates was determined to estimate the potential public health risk posed by the detected isolates.

## 2. Material and Methods

### 2.1. Sample Collection

In total, 33 samples were taken from great cormorants living at or close to the coastline of the North Sea (*n* = 13) or Baltic Sea (*n* = 20) of Schleswig-Holstein, Germany. Three colonies were sampled, one in the Wallnau Waterbird Reserve on the island of Fehmarn (Baltic Sea), one in the nature reserve Geltinger Birk (Baltic Sea), and one in Friedrichsgabekoog (North Sea), an agricultural landscape close to Büsum. A single sample was collected on the island of Sylt (North Sea). A map created with ArcGIS (ESRI, Redlands, CA, USA, Version 10.6.1) indicates all the sample locations (Figure 1). The samples were obtained either as cloacal or faecal swabs (Sterile transport swabs, Heinz Herenz, Germany). The cloacal swabs were taken during the bird ringing of nestlings. Therefore, the nestlings were carefully gripped out of their nests, fixed, equipped with a small metal tag, and then sampled by inserting a sterile swab into the cloaca and subsequently turning it several times on the mucosa. Faecal swabs were taken from fresh faeces from the ground under the nesting trees and one on the beach after witnessed defecation. Here, a sterile swab was inserted into the excrement and also turned a few times to gain enough material. All the swabs were taken between 28 May and 21 July 2017. The samples were transported to the lab immediately without refrigeration. 

### 2.2. E. coli Isolation and Identification

The obtained swabs were stored at room temperature for between three and six months. The sample processing aimed at the isolation of only the resistant *E. coli*. This was performed via selective enrichment for Enterobacteriaceae and following transfer to selective agar plates supplemented with antimicrobials. Mossel broth (Carl Roth GmbH & Co. KG, Karlsruhe, Germany), blood agar (Carl Roth GmbH & Co. KG), and Gassner agar (Sigma Aldrich Chemie GmbH, Steinheim am Albuch, Germany) were prepared in accordance with the manufacturer’s instructions. MacConkey agar (Carl Roth GmbH & Co. KG), which was additionally supplemented with different antimicrobials, was also used. The following antimicrobials and concentrations were used to supplement the agar plates: ampicillin (30 mg/L; Carl Roth GmbH & Co. KG), cephalothin (30 mg/L; TCI Europe N.V., Zwijndrecht, Belgium), chloramphenicol (10 mg/L; Carl Roth GmbH & Co. KG), ciprofloxacin (1 mg/L; Alfa Aeser, Thermo Fischer Scientific, Waltham, MA, USA), colistin (2 mg/L; AppliChem GmbH, Darmstadt, Germany), gentamicin (10 mg/L; Carl Roth GmbH & Co. KG), sulfamethoxazole (512 mg/L; Sigma Aldrich Chemie GmbH), and tetracycline (15 mg/L; AppliChem GmbH). Each swab was streaked on blood and Gassner agar plates and on Mossel broth and incubated for 24 h at 37 °C, to be used as a reference for the bacterial growth. In the following, 100 µL of the Mossel bouillon was streaked on each of the eight different antimicrobial-supplemented MacConkey agar plates. Again, the plates were incubated for 24 h at 37 °C. From every plate showing bacterial growth, five colonies were selected, and if less than five colonies were available, then all the colonies were selected. The selected colonies were then suspended in glycerine and stored at −80 °C. The colonies were selected based on their macroscopical appearance. On MacConkey agar, *E. coli* colonies (among others, e.g., *Klebsiella* spp. and *Enterobacter* spp.) are typically coloured red or pink with a bile salt precipitation halo surrounding the colony. Bacterial growth on the antimicrobial containing agar plates are listed in the Appendix A.

The identification of *E. coli* was performed using selective plating on Chromocult (Merck KGaA, Darmstadt, Germany), Gassner, and MacConkey agar plates. Due to their lactose fermentation ability, *E. coli* typically appear as blue to violet (Chromocult), blue (Gassner, Kempten, Germany), and red or pink colonies (MacConkey). Afterwards, the presumptive *E. coli* were confirmed via *gadA*-PCR [57]. The isolates that did not yield a positive PCR result were subsequently analysed by MALDI-TOF mass spectrometry for species identification. After this, 81 confirmed *E. coli* isolates were obtained. As each sample was streaked on eight different plates and up to five colonies per plate were collected, one sample delivered from one to nine isolates.

### 2.3. Resistance Phenotype

Diskdiffusion tests were performed in accordance with the Clinical and Laboratory Standards Institute standards [58] to determine the resistance profiles of the 81 confirmed *E. coli* isolates. Fourteen different antimicrobial-infused disks were purchased from Oxoid (Wesel, Germany). The tested antimicrobials comprised amoxicillin/clavulanic acid (20/10 µg), ampicillin (10 µg), cefazolin (30 µg), cefpodoxime (10 µg), chloramphenicol (30 µg), florfenicol (30 µg), streptomycin (10 µg), gentamicin (10 µg), kanamycin (30 µg), ciprofloxacin (5 µg), nalidixic acid (30 µg), compound sulfonamide (300 µg), trimethoprim (5 µg), and tetracycline (30 µg), representing seven different antimicrobial classes. For each isolate, a bacterial suspension in NaCl with 0.5 on the MacFarland scale, corresponding to approximately 10^8^ CFU/mL, was prepared and inoculated on the surface of Mueller–Hinton (Carl Roth GmbH & Co. KG) agar plates, after which a maximum of six antimicrobial disks were placed on the agar surface. The plates were then incubated for 18 h at 35 °C ± 2 °C. The inhibition zones were evaluated in accordance with the CLSI standards [59,60]. The intermediate susceptible isolates and resistant isolates were grouped together for subsequent analyses. The isolates suspected of being colistin-resistant were additionally tested using MacConkey agar plates supplemented with colistin (2 mg/L). As there are no CLSI breakpoints for colistin based on zone diameters, the colistin resistance of the isolates growing on these plates was confirmed using the broth macrodilution method, as described by the CLSI [58,59]. If multiple isolates were obtained from the same swab but showed the same resistance profile, only one was used for further testing to avoid copy isolates. Thirty-five isolates were identified as duplicates. One isolate did not show any resistance in the disk diffusion test. In the end, 45 *E. coli* isolates with phenotypic resistances were gained from the 33 samples. The 45 isolates were distributed among the sampling locations as follows: 15 from Wallnau, 9 from Geltinger Birk, 16 from Friedrichsgabekoog, and 5 from Sylt. Moreover, the isolates which showed a resistance profile to three or more antimicrobials with different action mechanisms were classified as multidrug-resistant.

### 2.4. Resistance Genotype

The isolates were tested for the presence of antimicrobial resistance genes using a panel of PCR assays, as described previously [61]. The targeted genes comprised *strA*, *strB* [62], *aadA1* [63], *aadA2* [64], *ant-(2″)-I*, *aac(3)-II*, and *aac(3)-IV*, mediating resistance to aminoglycosides; *bla*_TEM_, *bla*_SHV_, *bla*_OXA-1-like_, and *bla*_OXA-2_, mediating resistance to β-lactams; *aac(6′)-Ib-cr*, *qnrA*, *qnrB*, *qnrC*, *qnrD*, and *qnrS*, associated with reduced susceptibility to quinolones; *catA1*, *catA2*, *catA3*, *catB2*, *catB3*, *cmlA*, and *floR*, mediating resistance to phenicols; *sul1*, *sul2*, and *sul3*, mediating resistance to sulfonamides; *dfrA1/15/16*, *dfrA5/14*, *dfrA7/17*, and *dfrB1/2/3*, mediating resistance to trimethoprim; *tet*(A), *tet*(B), *tet*(C), *tet*(D), *tet*(E), *tet*(G), *tet*(H), *tet*(L) [65], *tet*(M), and *tet*(O), mediating resistance to tetracyclines [61]; and *mcr-1* [66], *mcr-2*, *mcr-3*, *mcr-4*, *mcr-5* [67], *mcr-6*, *mcr-7*, *mcr-8*, and *mcr-9* [68], mediating resistance to colistin. 

### 2.5. Molecular Typing

All of the *E. coli* isolates were assigned to one of the four major phylogenetic groups (A, B1, B2, and D) using PCR assays targeting two genes (*chuA* and *yjaA*) [69], as well as the anonymous DNA fragment TSPE4.C2 [57]. The isolates were further tested for the presence of five genes indicative of avian pathogenic *E. coli* (APEC): *iroN*, *ompT*, *hlyF*, *iss*, and *iutA* [70]. All the isolates were also subjected to XbaI macrorestriction analysis and the subsequent pulsed-field gel electrophoresis (PFGE), according to the published protocols [71]. An UPGMA analysis (dice coefficient, 0.5% optimization, and 1% position tolerance) was performed using the Bionumerics software (version 7.6, Applied Maths, Sint-Martens-Latem, Belgium). Band patterns are shown in the Appendix A.

### 2.6. Statistical Analysis

Statistical analyses were performed using RStudio (RStudio PBC, Boston, MA, USA) version 1.4.1103 (R version 4.1.2) with the packages readxl and DescTools. Due to the small sample size, a Fisher’s exact test was performed to test the significance of the association between (i) the number of samples with resistant *E. coli* isolates and the sampling locations and (ii) the number of multidrug-resistant isolates (including only samples with resistant *E. coli*) and the sampling locations. The level of significance was set at *p* < 0.05.

## 3. Results

### 3.1. Occurrence of Antimicrobial-Resistant E. coli Isolates

Of the 33 great cormorant samples that were collected for this study, 87.9% (29/33) had bacterial growth on at least one of the antimicrobial-containing agar plates. In 66.7% (22/33) of the samples, at least one resistant *E. coli* isolate was obtained, while the other isolates could not be confirmed as *E. coli*. Resistant *E. coli* were present in 14 of the 20 (70%) samples from the Baltic Sea coast, including 9 of the 11 (81.8%) samples from Wallnau and 5 of the 9 (55.6%) samples from Geltinger Birk. Regarding the samples collected at the North Sea coast, 8 of the 13 (61.5%) samples contained resistant *E. coli*, including 7 of the 12 (58.3%) samples collected in Friedrichsgabekoog, as well as the single sample (100%) from Sylt. No statistically significant association between the number of samples with resistant isolates and the sampling locations (*p* = 0.5393) was observed. Whether there were differences between nestlings and adult birds could not be calculated, because the faecal samples could also have come either from nestlings or adults. The number and the percentage of the samples containing resistant *E. coli* isolates, as well as their allocation for the different locations, are depicted in Figure 2. In total, 45 different resistant *E. coli* isolates were collected and further investigated (Figure 3).

### 3.2. Resistance Pheno- and Genotypes

Overall, most of the 45 *E. coli* isolates (36/45, 80%) showed resistance to streptomycin in the diskdiffusion tests, followed by resistance to tetracycline (25/45, 55.6%) and sulfonamides (24/45, 53.3%). Resistance to amoxicillin/clavulanic acid, ampicillin, and trimethoprim ranged between 35.6 and 31.1% (14, 15 and 16/45), followed by cefazolin (13/45, 28.9%), kanamycin (10/45, 22.2%), ciprofloxacin and florfenicol (6/45, 13.3% each), and chloramphenicol and nalidixic acid (5/45, 11.1% each). The lowest rates of resistance were seen for gentamicin (4/45, 8.9%) and colistin (1/45, 2.2%). None of the isolates showed resistance against cefpodoxime.

The phenotypic resistance patterns varied in the isolates from the different sampling locations. In the isolates from the samples of Geltinger Birk (nature reserve) and Friedrichsgabekoog (agricultural landscape), the most common resistance phenotype was to streptomycin, with percentages of 77.8 (7/9) and 93.8 (15/16), respectively, while in Wallnau (waterbird reserve) most of the isolates were resistant to sulfonamides (73.3%, 11/15). Four out of the five (80%) isolates recovered from the sample taken in Sylt showed resistance to streptomycin, ampicillin, and cefazolin. Apart from the zero isolates with resistance to cefpodoxim, the isolates from the samples collected in Geltinger Birk were all susceptible to chloramphenicol; the isolates from Wallnau and Sylt were all susceptible to ciprofloxacin and nalidixic acid; and the isolates from Geltinger Birk and Sylt were all susceptible to gentamicin. Only one isolate from Friedrichsgaabekoog was resistant to colistin. The resistance patterns of the isolates from the four locations to the different antimicrobials are shown in Figure 4.

Of the 45 isolates, 10 isolates were resistant to only one antimicrobial agent (22.2%), and 7 isolates showed resistance to two antimicrobials (15.6%). Resistance to three antimicrobials was shown by 9 (20%) isolates; resistance to four and five antimicrobials was shown by 5 isolates for each (each 11.1%); resistance to six antimicrobials was shown by 4 isolates (8.9%); and resistance to seven antimicrobials was shown by 1 isolate (2.2%). Three isolates (6.7%) were resistant to 10 antimicrobials, and one isolate (2.2%) was resistant to 11 of the tested antimicrobials. 

Considering the different antimicrobial classes instead of the individual antimicrobial agents revealed that the isolates showed the highest resistance rates to aminoglycosides (here represented by streptomycin, gentamicin, and kanamycin) (37/45, 82.2%), followed by resistance to tetracyclines (here represented by tetracycline) (25/45, 55.6%), folate synthesis inhibitors (here represented by sulfonamides and trimethoprim) (24/45, 53.3%), and non-extended spectrum β-lactams (here represented by amoxicillin/clavulanic acid, ampicillin, and cefazolin) (22/45, 48.9%). Lower resistance rates occurred towards quinolones (here represented by ciprofloxacin and nalidixic acid), phenicols (here represented by chloramphenicol and florfenicol) (6/45, 13.3% each), and polymyxins (here represented by colistin) (1/45, 2.2%). No extended spectrum β-lactamases (here represented by cefpodoxime) were detected. Resistance to at least one antimicrobial agent of three or more antimicrobial classes was rated as multidrug resistance, which occurred in 48.9% (22/45) of the *E. coli* isolates. Of these, eleven were resistant to three antimicrobial classes, six to four classes, three to five classes, and two isolates to six antimicrobial classes. Resistance to two antimicrobial classes occurred in 31.1% (14/22) of the isolates and to one antimicrobial class in 20% (9/45). Of the different locations, the highest rate of multiresistance was found in the isolates from Friedrichsgabekoog (9/16, 56.3%), followed by Wallnau (8/15, 53.3%), Sylt (2/5, 40%) and Geltinger Birk (3/9, 33.3%). The number of susceptible and resistant isolates per location, as well as their proportion of multiresistance, is depicted in Figure 5. No statistically significant association between the number of samples containing multidrug-resistant *E. coli* and the sampling locations (*p* = 0.924) was observed.

Isolates showing phenotypic resistance were tested for a selection of corresponding resistance genes. By total numbers, the resistance genes most commonly detected were *sul2* (*n* = 20) and *sul1* (*n* = 14) mediating resistance to sulfonamides; *strA* (*n* = 18), *strB* (*n* = 18), and *aadA1* (*n* = 11) mediating resistance to streptomycin; and *tet*(A) (*n* = 16) and *tet*(B) (*n* = 10) mediating resistance to tetracyclines, as well as *bla_TEM_* (*n* = 15) mediating resistance to non-extended spectrum β-lactams. More rarely found resistance genes include *dfrA5/A14* (*n* = 6), *dfrA7/A17* (*n* = 3), and *dfrA1/15/16* (*n* = 5) mediating resistance to trimethoprim; *floR* (*n* = 5) mediating resistance to chloramphenicol; *aac(3)-II* (*n* = 1) and *ant(2“)-I* (*n* = 2) mediating resistance to aminoglycosides; *tet*(D) (*n* = 1) mediating resistance to tetracyclines; *sul3* (*n* = 1) mediating resistance to sulfonamides; and *qnrS* (*n* = 2) mediating resistance to quinolones.

Of the isolates showing phenotypic resistance to aminoglycosides (*n* = 37), 25 (67.6%) carried one to four corresponding resistance genes that were tested here, whereas all of the isolates resistant to tetracycline (*n* = 25) carried one to two corresponding resistance genes. Of the isolates with resistance to sulfonamides (*n* = 24), in 23 (95.8%) one to two corresponding resistance genes were detected, while all the isolates resistant to trimethoprim (*n* = 14) carried one corresponding resistance gene. Of the isolates showing resistance to β-lactams (*n* = 22), 15 (68.2%) carried one resistance gene, while of the isolates showing resistance to phenicols (*n* = 6), in 5 (83.3%) one corresponding resistance gene was detected. Of the isolates showing resistance to quinolones (*n* = 6), two carried (33.3%) one corresponding gene, and the isolate showing resistance to colistin did not carry any of the tested resistance genes. Of the 26 isolates that exhibited phenotypic resistance or intermediate resistance, but lacked a corresponding resistance gene, 21 (80.8%) exhibited intermediate resistance. Only in a single isolate with phenotypic resistance to cefazolin could no corresponding resistance gene tested be detected here. However, from the isolates without a detected plasmid-located quinolone resistance gene, three out of four showed a full phenotypic resistance. The detected resistance genes and the number of genes per location are summarized in Table 1. For each isolate, the resistance genotypes are shown in Figure 3.

### 3.3. Molecular Typing

The determination of major phylogenetic groups assigned most of the isolates to the phylogenetic group B1 (23/45, 51.1%), followed by the phylogenetic group D (13/45, 28.9%), the phylogenetic group B2 (5/45, 11.1%), and the phylogenetic group A (4/45, 8.9%). The phylogenetic groups of each isolate are listed in Figure 3. The results for the different sampling locations, as well as for all the isolates, are summarized in Table 2.

Eleven isolates were identified as avian pathogenic *E. coli* (APEC) isolates (all five indicative genes attested) while the remaining 34 isolates were presumed to be commensal *E. coli* isolates. Of the different locations, the one sample from Sylt had the highest rate of APEC isolates (2/5, 40%), followed by the samples from Wallnau (5/15, 33.3%) and Friedrichsgabekoog (4/16, 25%). No APEC isolates were detected in the samples from Geltinger Birk. The results of the analysis for the APEC isolates identification are shown in Figure 3.

The majority of the APEC isolates (9/11) were multidrug-resistant. With regard to the phylogenetic groups, 45.5% (5/11) of the APEC isolates belong to group B1, 36.4% (4/11) to group D, and 18.2% (2/11) to group A. None of the APEC isolates belonged to group B2. The phylogenetic group B1 isolates also had the highest proportion of multidrug resistance, with 45.5% (10/22), while 31.8% (7/22) belonged to group D, 13.6% (3/22) to group A, and 9.1% (2/22) to group B2.

Macrorestriction analyses revealed band patterns in 42 isolates, based on which the genomic relatedness could be determined. The remaining three isolates were found to be non-typeable by Xbal macrorestriction. It was possible to identify one cluster of three isolates with ≥90% similarity, involving two isolates from the same sample and one isolate from a different sample of the same sampling location and sampling date (isolates 25a, 25b and 26b), with two isolates having identical band patterns (25a and 25b). Another cluster with ≥80% similarity included three isolates of different samples of the same sampling location and sampling date (isolates 8b, 9b and 7a), with two isolates (8b and 9b) showing ≥90% similarity. Additionally, two isolates from three samples showed ≥90% similarity, with two isolates from the same sampling date and location having identical band patterns. All the other isolates showed a high degree of heterogeneity. The dendrogram of the 41 isolates is depicted in Figure 3.

## 4. Discussion

In the present study, antimicrobial-resistant *E. coli* isolates were detected in 66.7% of great cormorant samples collected at four different sites in the federal state of Schleswig-Holstein, Germany. For the three sampled great cormorant colonies, the occurrence varied between 55.6, 58.3, and 81.8%, respectively. Only a few other studies have investigated antimicrobial-resistant *E. coli* exclusively in cormorant species. In contrast to the present study, *E. coli* were isolated and then tested for their antimicrobial susceptibility, and very low resistance rates were found. In Japan, only 3.5% of *E. coli* isolated from great cormorants presented an antimicrobial-resistant profile [53]. A similarly low rate of 4.8% resistance was found in double-crested cormorant (*Phalacrocorax auritus*) chicks in Canada [72]. On the other hand, there are numerous reports of isolated antimicrobial-resistant *E. coli* in a multitude of wild bird species, but only a few of them found similar or higher rates of samples carrying antimicrobial resistance than were ascertained in this study. One study reported resistance rates in over 90% of the samples [73], while a few others reported around 50 [47,74,75] or 30% [76,77,78,79], respectively, of the samples carrying resistant *E. coli*. Lower rates varied from 1.4 to 18% [44,45,79,80]. The reasons for the wide diversity of bird species carrying resistant isolates may include bird population density [31], the feeding ecology of the birds [81], the close or absent proximity to livestock farms or wastewater run-off [82], and human population density in the sampling areas [83,84], as well as a seasonality depending on temperature [85]. It is noteworthy that differences in the study design, such as the selection of bird species (feeding ecology), the different status of the sampled animals (live/deceased, likely healthy/weak or sick), and the sampling method (throat/cloacal/faecal swab or organ samples), as well as the sample processing (only one isolate per sample/several isolates per sample), make the comparison of the results difficult. Furthermore, due to the delay in sample processing in the present study, it was of course possible that the number of *E. coli* was underestimated.

Multidrug resistance, defined as a resistance against three or more antimicrobial classes [86], occurred in the present study in 48.9% of the resistant *E. coli* isolates. Compared to other studies, this is a high rate. However, comparing studies is limited as the definition of multidrug resistance varies. Applying the same definition of multidrug resistance as in the present study, samples from different bird species collected in Poland carried multidrug-resistant *E. coli* in 38.6% [47] and 31.2% [73] of the cases, respectively. Furthermore, 15.3% of the samples of migratory birds collected in China [32], as well as 6.7% of the great cormorants samples collected in Switzerland [54] had multidrug-resistant *E. coli* isolates. Multidrug resistance, defined as resistance against two or more antimicrobial classes, was found in suburban and in wild birds in Australia in 2.1% and 1.5% of the tested samples, respectively [45]. Resistance against three or more antimicrobials was reported in 3% of adult gulls in Alaska [75] and in 3.2% of different bird species in Germany [44]. Resistance against two or more antimicrobials was detected in 19% of gulls tested in different European countries [77] and in 3.2% of the double-crested cormorant chicks in Prince Edward Island, Canada [72].

The high rate of multidrug resistance in the present study seems to indicate a high contamination of the environment by multidrug-resistant bacteria [39]. The highest amount of multidrug-resistant isolates was found in the great cormorant colony in Friedrichsgabekoog, while the location with the highest percentage of isolates with resistance to at least one antimicrobial was Wallnau. However, the highest human population densities are in Wallnau (Ost-Holstein, Germany), with 145 humans/km^2^, while Friedrichsgabekoog (Dithmarschen, Germany) and Geltinger Birk (Schleswig-Flensburg, Germany) are less populated, with 94 humans/km^2^ and 98 humans/km^2^, respectively [87]. Of the three colony sampling locations, Wallnau, a waterbird reserve, and Geltinger Birk, a nature reserve, may be seen as more pristine than Friedrichsgabekoog, which is an agricultural environment. Moreover, differences in the sampling method need to be noted. Only in Wallnau were cloacal swab samples from nestlings taken, while in the other two colonies the samples were obtained by swabbing fresh faeces collected from the ground, comprising a higher rate of contamination. In the latter two colonies, no information is available on the age class of the sampled birds. The faeces could have come from nestlings or adults. Although other studies found a higher occurrence of resistant bacteria in areas with a higher anthropogenic impact [45,82,83], the results of the different sampling locations in the present study did not show statistically significant differences.

Looking at the antimicrobial classes, most of the isolates in the present study showed resistance to aminoglycosides (82.2%), followed by tetracyclines (55.6%), folate synthesis inhibitors (53.3%), and non-extended spectrum β-lactams (48.9%). In most studies investigating antimicrobial-resistant *E. coli* in wild birds, tetracycline resistance was the most common of all the antimicrobials tested [47,72,73,76,77,78]. In other studies, ampicillin or β-lactams were named as antimicrobials with the highest rate of resistant isolates [32,47,74], while in one study this was the case for sulfadimethoxine [44]. In the present study, streptomycin resistance occurred most frequently (80%). This is of specific concern as streptomycin is listed as a watch group antibiotic in the WHO Access, Watch, Reserve (AWaRe) classification of antibiotics for evaluation and monitoring of use, indicating that it is as an essential first or second choice for empiric treatment of a limited number of specific infectious syndromes [88]. In other studies, streptomycin resistance varied from 47% [44], to 32.1% [74], and to 9.3% [78]. Again, it is difficult to compare the different studies as the tested antimicrobials were not consistent.

The findings in the present study are not in line with the antimicrobial consumption rates in Germany. Between 2011 and 2014, the amounts of antimicrobials per antimicrobial class in tons, including only those tested in this study, sold to Germany-based veterinarians had the following order: penicillins, tetracyclines, sulfonamides, aminoglycosides, folate synthesis inhibitors, fluoroquinolones, cephalosporins, and phenicols [89]. In the same period, β-lactams and fluoroquinolones were the most frequently prescribed antimicrobials in hospitals, while for outpatients β-lactams and tetracyclines had the highest prescription rates [89]. If the consumption rate alone was responsible for the amount of antimicrobial resistance and the contamination of the environment, β-lactams would have been the antimicrobials with the highest resistance rates in the present study. Some studies also found extended-spectrum beta-lactamases (ESBL) producing *E. coli* in wild birds [32,52,54,79,90], which was not the case in the present study.

In the present study, the resistance genes most commonly detected with regard to the total numbers were *sul2* and *sul1* (mediating resistance to sulfonamides); *strA*, *strB*, and *aadA1* (mediating resistance to aminoglycosides); and *tet*(A) and *tet*(B) (mediating resistance to tetracyclines), as well as *bla*_TEM_ (mediating resistance to non-extended spectrum β-lactams). More rarely found resistance genes include aa*c(3)-II* and *ant(2″)-I* (mediating resistance to gentamicin); *floR* (mediating resistance to chloramphenicol); *tet*(D) (mediating resistance to tetracyclines); *sul3* (mediating resistance to sulfonamides); *dfrA5/A14*, *dfrA7/A17*, and *dfrA1/15/16* (mediating resistance to trimethoprim); and *qnrS* (mediating resistance to quinolones). Some of the genes detected in the present study were also reported in *E. coli* isolated from different wild bird species, including *bla*_TEM_, *sul2*, *sul3*, *tet*(A), *strA*, *strB*, and *aadA* [73], *bla*_TEM_, *aadA*, *aac(3)-II*, *aac(3)-IV*, *sul1*, *sul2*, and *sul3* [74], and *bla*_TEM-1_, *tet*(A), *tet*(B), *tet*(M), *sul1*, *sul2*, *sul3*, and *floR* [32]. A corresponding resistance gene was not detectable for all resistant isolates. In the present study, 26 phenotypically resistant isolates (5 resistant, 21 intermediate) did not carry one of the tested resistance genes. This can be explained by the fact that only a selection of the resistance genes was tested, which is especially true for aminoglycosides [91]. Furthermore, the phenotypic resistance may be caused by a chromosomal mutation rather than by the dedicated resistance genes which we tested here. This is especially known for quinolone and colistin resistances [92,93,94,95]. In addition, other mechanisms such as chromosomally encoded efflux pumps can be involved in the formation of resistance [96,97].

To assess the pathogenetic potential of the isolated *E. coli* of the present study, their affiliation to phylogenetic groups and avian pathogenic *E. coli* (APEC) strains was determined. The assignment to phylogenetic groups is primarily performed to classify isolates with regard to their phylogenesis. In addition, this classification is used to approximately assign isolates to be either commensals or pathogens. In this regard, the isolates belonging to phylogenetic group A are considered to be mainly commensals; the isolates of group B1 may more often include pathogens, while the ones belonging to group D and especially group B2 are more likely to carry genes associated with extraintestinal pathogenicity to humans [98,99,100,101]. In the present study, the isolated *E. coli* belonged to 60% of the phylogenetic groups A (8.9%) and B1 (51.1%), thus being more likely to represent commensals. The remaining 40% have a higher potential to be pathogenic, belonging to the groups B2 (11.1%) and D (28.9%). Similarly to this, other studies on wild birds also found that more than 50% of the *E. coli* isolates belonged to the phylogenetic groups most likely harbouring commensals [40,44,46,73,74,102], leaving the proportion of potential pathogenic *E. coli* isolates over all listed studies between 10.4 to 44.5%. In the present study, the isolates belonging to phylogenetic groups less frequently associated with human-pathogenic isolates showed a higher proportion of multidrug resistance, with 59.1% of all the multidrug-resistant isolates belonging to group B1 (45.5%) or A (13.6%). Accordingly, only 40.9% belonged to group B2 (9.1%) or D (31.8%), with the lowest rates of multiresistance found in the isolates belonging to group B2. In contrast, another study found that 66.7% of multidrug-resistant *E. coli* isolates from different German bird species belonged to the phylogenetic groups B2 (46.7%) and D (20%) [44].

With regard to the APEC assignment, 24.4% of the resistant isolates of the present study were classified as APEC. Interestingly, APEC isolates belonged mainly to the phylogenetic groups harbouring potential commensals (18.2% to group A and 45.5% to group B1). Consistently with the results presented here, different studies have reported that *E. coli* of animal origin often noticeably belong to group B1, indicating a potential host-adaptation [103,104]. None of the APEC isolates belonged to group B2, although the isolates of this phylogenetic group are considered most likely to harbour pathogenic strains [98,100], which indicates that the classification into commensals and pathogens via phylogenetic grouping is not strict. On the other hand, most of the APEC isolates (81.8%) were multidrug-resistant. This fact is especially alarming as multidrug-resistant pathogenic *E. coli* transmitted, e.g., via natural waterbodies, to humans, companion animals, or livestock could cause severe disease with limited antimicrobial therapy possibilities. Thus, monitoring environmental contamination with such microorganisms is of specific importance under the one health perspective.

## 5. Conclusions

The present study shows that great cormorants, as has been shown for many other wild bird species, can carry antimicrobial-resistant *E. coli*, although wild birds are generally not subjected to antimicrobial treatment themselves. Simultaneously, they can act as vectors by shedding antimicrobial-resistant bacteria into the environment. As great cormorants are almost strictly piscivorous, they can be used as important sentinels for environmental contamination and as effective bioindicators. Of specific concern is the high number of multidrug-resistant *E. coli* isolates detected on the North and Baltic Sea coasts of Schleswig-Holstein. This environmental contamination poses a potential hazard to human, companion animal, and livestock health, as infections or colonization with bacteria are possible for all of those coming in close contact with the correlating environmental sources. Furthermore, regular surveillance of sentinel species is essential to determine variances in the degree of antimicrobial contamination of the environment and may help in assessing the effectiveness of mitigation measures against the global problem of antimicrobial resistance.

## Figures and Tables

**Figure 1 pathogens-11-00836-f001:**
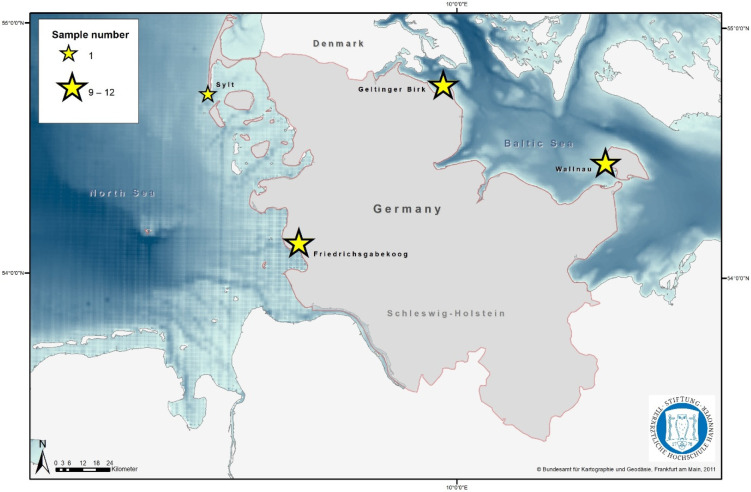
Sample map. The map indicates the four sampling locations, Wallnau, Geltinger Birk, Friedrichsgabekoog, and Sylt, as well as the sample size.

**Figure 2 pathogens-11-00836-f002:**
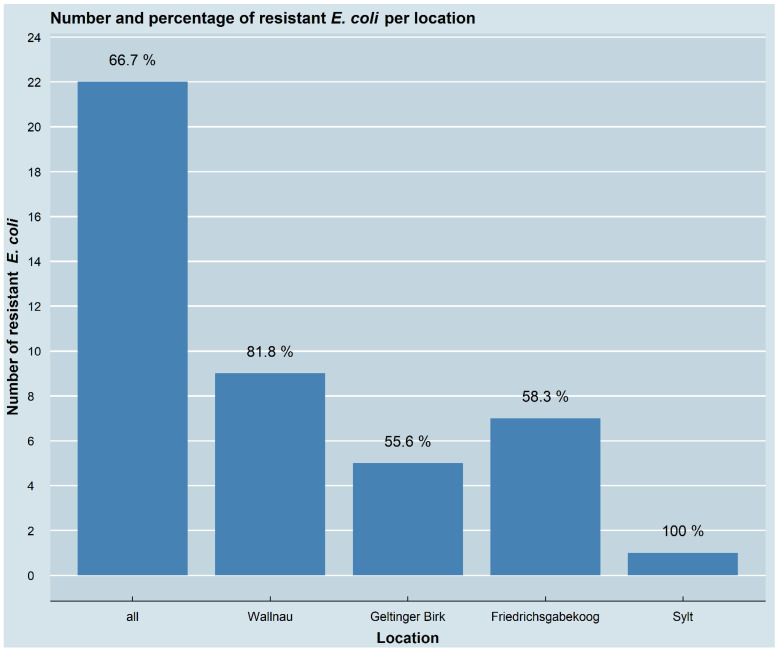
Number and percentage of resistant *E. coli* per location. Shown are all samples (*n* = 22) with at least one resistant *E. coli* isolate as well as their distribution in the different sampling locations. Percentages indicate proportion of samples with resistant *E. coli* isolates to the total number of samples for all locations together, as well as for each location separately.

**Figure 3 pathogens-11-00836-f003:**
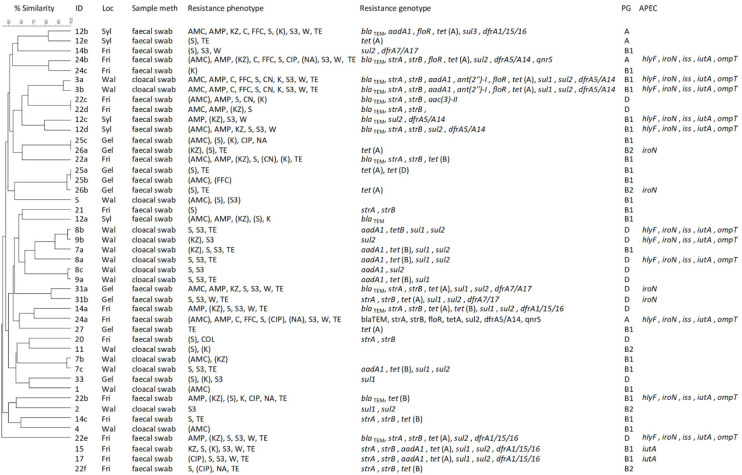
Overview of the pheno- and genotypes of *E. coli* isolates from wild cormorants. For 42 isolates the dendrogram is shown; three isolates were not typeable. For all isolates, the sampling location (Loc), the sample method, the resistance pheno- and genotypes, and the phylogenetic group (PG), as well as the detection of genes indicative for avian pathogenic *E. coli* (APEC), are listed. Locations: Wal = Wallnau, Gel = Geltinger Birk, Fri = Friedrichsgabekoog, Syl = Sylt. Resistance phenotype: AMC = amoxicillin/clavulanic acid, AMP = ampicillin, KZ = cefazolin, CPD = cefpodoxime, C = chloramphenicol, FFC = florfenicol, S =streptomycin, CN = gentamicin, K = kanamycin, CIP = ciprofloxacin, NA = nalidixic acid, S3 = compound sulfonamide, W = trimethoprim, TE = tetracycline, and COL = colistin.

**Figure 4 pathogens-11-00836-f004:**
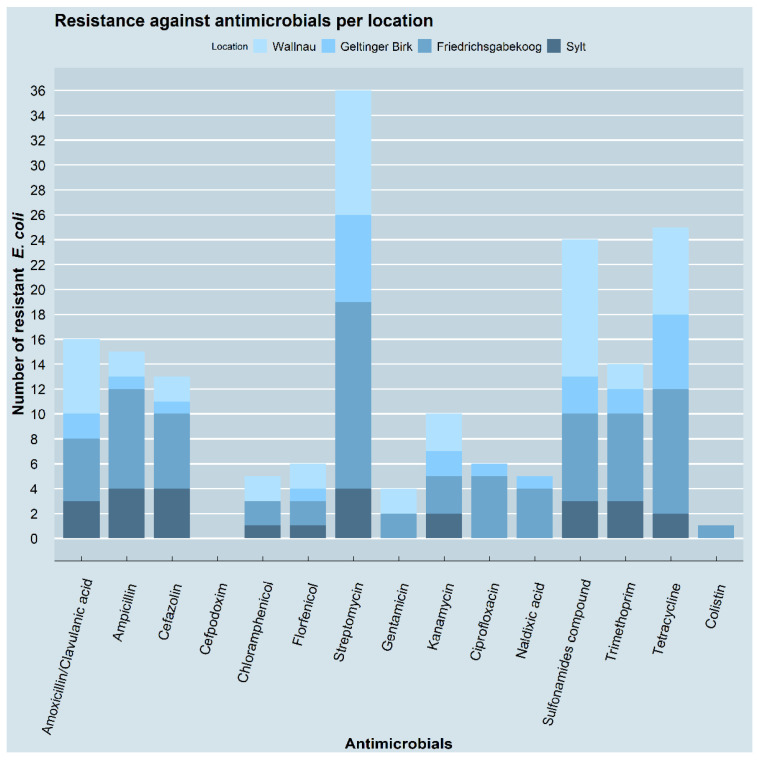
Resistance against antimicrobials per location. The number of resistant *E. coli* isolates is shown for each of the tested antimicrobials. The proportion of the four locations is indicated by different colours. Isolates with more than one resistance are included multiple times. The figure was created using RStudio version 1.4.1103 (R version 4.1.2) with the package ggplot2.

**Figure 5 pathogens-11-00836-f005:**
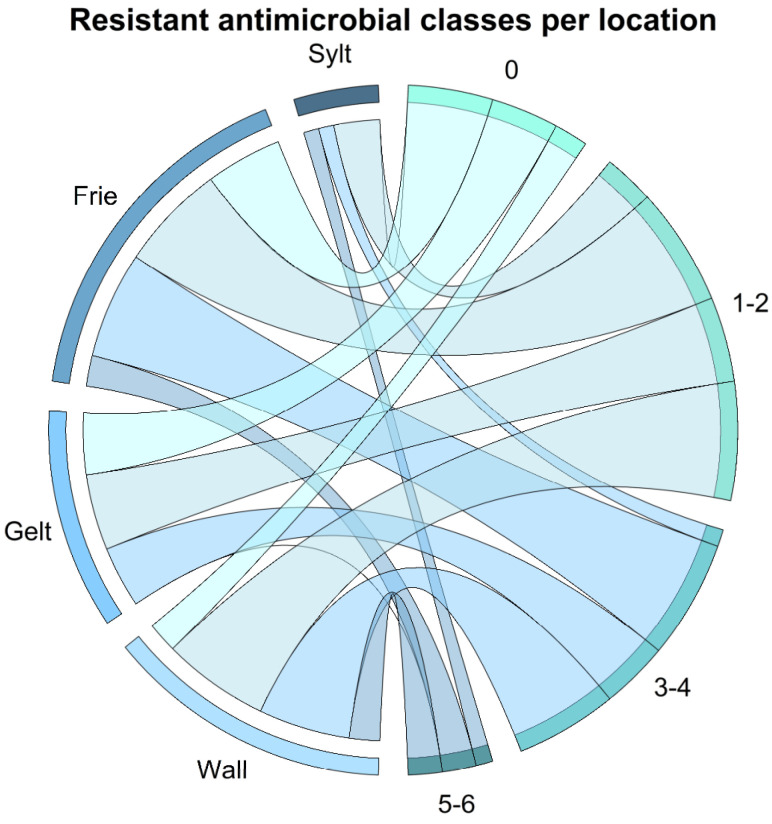
Resistance to antimicrobial classes per location. For each location the number of resistant isolates plus the number of samples without resistance are indicated on the left. Straps show the proportion of the number of isolates resistant against different antimicrobial classes. Wall = Wallnau, Gelt = Geltinger Birk, Frie = Friedrichsgabekoog, 0 = sample without resistance, 1–2 = isolates with resistance against one or two antimicrobial classes, 3–4 = isolates with resistance against three or four antimicrobial classes, 5–6 = isolates with resistance against five or six antimicrobial classes. Multiresistant isolates are resistant against at least one antimicrobial from three or more antimicrobial classes. The figure was created using RStudio version 1.4.1103 (R version 4.1.2) with the packages readxl and DescTools.

**Table 1 pathogens-11-00836-t001:** Resistance genes—list of detected resistance genes as well as number of resistance genes per location and over all locations. Wall = Wallnau, Gelt = Geltinger Birk, Frie = Friedrichsgabekoog.

Resistance Gene	Wall	Gelt	Frie	Sylt	Total
*bla* _TEM_	2	1	8	4	15
*strA*	2	2	13	1	18
*strB*	2	2	13	1	18
*aadA1*	8	0	2	1	11
*aac(3)-II*	0	0	1	0	1
*ant(2“)-I*	2	0	0	0	2
*floR*	2	0	2	1	5
*tet*(A)	2	6	6	2	16
*tet*(B)	5	0	5	0	10
*tet*(D)	0	1	0	0	1
*sul1*	8	3	3	0	14
*sul2*	9	2	7	2	20
*sul3*	0	0	0	1	1
*dfrA5/A14*	2	0	2	2	6
*dfrA7/A17*	0	2	1	0	3
*dfrA1/15/16*	0	0	4	1	5
*qnrS*	0	0	2	0	2

**Table 2 pathogens-11-00836-t002:** Phylogenetic groups—number and percentage of phylogenetic groups of isolates from the four sampling locations, as well as for all samples.

Phylogenetic Group	A	B1	B2	D
Wallnau	0	8 (53.3%)	2 (13.3)	5 (33.3)
Geltinger Birk	0	4 (44.4%)	2 (22.2%)	3 (33.3%)
Friedrichsgabekoog	2 (12.5%)	8 (50%)	1 (6.3%)	5 (31.3%)
Sylt	2 (40%)	3 (60%)	0	0
total	4 (8.9%)	23 (51.1%)	5 (11.1%)	13 (28.9%)

## Data Availability

The data presented in this study are available in this published article and its Appendix A.

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
