# Peer review of "High Rates of Multidrug-Resistant Escherichia coli in Great Cormorants (Phalacrocorax carbo) of the German Baltic and North Sea Coasts: Indication of Environmental Contamination and a Potential Public Health Risk"

_pathogens, 2022, doi:10.3390/pathogens11080836_

Round 1

Reviewer 1 Report

This paper is based on a low number of samples (of different kind) collected from several locations and therefor only limited conclusions can be made. It is interesting that a strict piscivore species harbor such a high frequency of MDR. It is recommended that the authors substantially reduce the text and focus on these core results. 

In the following there are some mor specific comments:

L 50: replace “organisms” with “bacteria”

L 56 – 58: When reading the references 21, 22 and 23: it becomes obvious that the authors miss/over-interpret these ones – especially the two last ones that are review papers with free, speculative reasoning. Pleas be more precise in citing other papers.

L 108 -111: Did you find any difference between these two types of samples; i.e. nestlings and adults? They have been exposed to microbes for different time periods. But of course a possible difference could be confounded by the geography.

L120: Did you really stored the swabs in RT for up to six months?

L119- 170: could you please make it clear for the reader why, give the rationale, you have two procedures (section 2.2 and section 2.3) for phenotypically determine resistance. Currently this is not justified.

L 201: please don’t use the term “prevalence” for a study with this design. And be consistent in writing “rate” or “frequency” to ease the understanding of the paper. Not range or other terms.

It is not obvious/explained for the reader how the 33 samples were translated into 45 isolates and is then the geographic/sampling site reasoning valid at all. Please clarify!

Over all, as it doesn’t seem to be any statistical differences between the sampling locations – this aspect of the study should be reduced to a minimum in the result section and discussion section.

L 377-403: This part of the discussion is very informative but suites better in a review paper – please focus and shorten substantially.

L417: it is common to use the term MDR if resistance occurs to three classes of antibiotics ii the same bacteria

L 505: The reasoning about the APEC isolates is confusing: they should by definition be pathogenic but it is stated that some of them were commensal? Please explain

L527; you mean livestock/domestic animal health? Please make clear as this is a “wildlife” paper.

L 528—538: please be more to the point in the conclusion. This is a very general/vague statement.

Reviewer 2 Report

This paper describes the occurrence of antimicrobial resistant Escherichia coli in free-living great cormorants of the North and Baltic Sea coasts of Schleswig-Holstein, GermanyThe manuscript is well written and easy to follow. The data were well analyzed and interpreted appropriately. High rates of multidrug resistant Escherichia coli were reported in this study. These findings are important for publick health surveillance.

Only one isolate showed resistance to colisin and none of the mcr genes were detected from this isolate. Just out of curiosity, Do you have any explanations for this? Could it be mcr-10 or a new variant of the mcr gene?

Minor comments

1. Line 181: there is a typo "mcr-1 [60], mcr-2, mcr-2...".

2. Figure 3: higher resolution would be preferred.

3. Line 323: there is an error with reference source.

4. Line 343: another error.

Round 2

Reviewer 1 Report

The manuscript is improved after the revision and provides some interesting findings.